# Development and characterization of microsatellite markers for population genetics of the cocoa pod borer *Conopomorpha cramerella* (Snellen) (Lepidoptera: Gracillaridae)

**Marynold Purificacion**[1☯¤], **Roslina Binti Mohd Shah**[2☯], **Thierry De Meeûs**[3,4], **Saripah Binti Bakar**[5], **Anisah Bintil Savantil**[2], **Meriam Mohd Yusof**[2], **Divina Amalin**[1], **Hien Nguyen**[6], **Endang Sulistyowati**[7], **Aris Budiman**[7], **Arni Ekayanti**[8], **Jerome Niogret**[9], **Sophie Ravel**[3], **Marc J. B. Vreysen**[10], **Adly M. M. Abd-Alla**[10]*

1 Biological Control Research Unit, Center for Natural Science and Environmental Research, De La Salle University, Manila, Philippines, 2 Centre for Cocoa Biotechnology Research, Malaysian Cocoa Board, Kota Kinabalu, Sabah, Malaysia, 3 Intertryp, Univ Montpellier, Cirad, IRD, Montpellier, France, 4 IRD, UMR Intertryp, Cirad, Campus International de Baillarguet, Montpellier, France, 5 Lembaga Koko Malaysia, Bagan Datuk, Perak, Malaysia, 6 Plant Protection Research Institute, Duc Thang, Hanoi, Vietnam, 7 Indonesian Coffee and Cocoa Research Institute, Jember, Indonesia, 8 Mars Cocoa Research Centre, Mars Wrigley, Sulawesi Selatan, Indonesia, 9 Mars Wrigley, Centre for Tropical Environmental & Sustainability Science, James Cook University Nguma-bada Campus, Smithfield, Australia, 10 Insect Pest Control Laboratory, Joint FAO/IAEA Centre of Nuclear Techniques in Food and Agriculture, Vienna, Austria

☯ These authors contributed equally to this work.
¤ Current address: Institute of Plant Breeding, College of Agriculture and Food Science, University of the Philippines Los Baños, Laguna, Philippines
* a.m.m.abd-alla@iaea.org

**Data Availability Statement:** All relevant data are within the manuscript and its Supporting

## Abstract

The cocoa pod borer (CPB) *Conopomorpha cramerella* (Snellen) (Lepidoptera: Gracillaridae) is one of the major constraints for cocoa production in South East Asia. In addition to cultural and chemical control methods, autocidal control tactics such as the Sterile Insect Technique (SIT) could be an efficient addition to the currently control strategy, however SIT implementation will depend on the population genetics of the targeted pest. The aim of the present work was to search for suitable microsatellite loci in the genome of CPB that is partially sequenced. Twelve microsatellites were initially selected and used to analyze moths collected from Indonesia, Malaysia, and the Philippines. A quality control verification process was carried out and seven microsatellites found to be suitable and efficient to distinguish differences between CPB populations from different locations. The selected microsatellites were also tested against a closely related species, i.e. the lychee fruit borer *Conopomorpha sinensis* (LFB) from Vietnam and eight loci were found to be suitable. The availability of these novel microsatellite loci will provide useful tools for the analysis of the population genetics and gene flow of these pests, to select suitable CPB strains to implement the SIT.

Information files. In addition the raw data are available in this link https://dataverse.harvard.edu/dataset.xhtml?persistentId=doi:10.7910/DVN/CNRSU8.

**Funding:** This work was funded by the International Atomic Energy Agency under the TC project RAS5095 and by the regular budget of the Joint FAO/IAEA Centre of Nuclear Techniques in Food and Agriculture. The funders had no role in study design, data collection and analysis, decision to publish, or preparation of the manuscript.

**Competing interests:** The authors have declared that no competing interests exist.

## Introduction

The cacao tree, *Theobroma cacao* L., belongs to the Sterculiaceae family and is one of the most widely cultivated species among the twenty-two species of the genus [1]. The International Cocoa Organization (ICCO) estimates the global market value of cocoa between 8 to 10 billion USD, based on an annual production of 4 million tons and a monthly average daily price of cocoa beans between USD 2,264 to 2,359 per ton. With the demand for finished and semi-finished cocoa products increasing yearly, cocoa production should ideally increase proportionally. However, the current worldwide production by the cocoa industry is not in line with this prediction. Farmers are producing less cocoa than world consumption, which inevitably leads to a chocolate deficit [2, 3]. The cocoa pod borer (CPB) *Conopomorpha cramerella* (Snellen) (Lepidoptera: Gracillaridae) is the major insect pest of cocoa production in South East Asia and it infests rambutan (*Nephelium lappaceum* L.), langsat (*Lansium parasiticum* (Osbeck) Sahni & Bennet) and cocoa [4]. CPB adults oviposit on the cocoa pods and the larvae enter the pods to feed on the pulp and placenta surrounding the beans that are used for chocolate production. On average, 40–60% of the cocoa beans produced in Southeast Asia are lost due to CPB and up to 80% of the crop is lost in farms in the absence of CPB management [5]. In addition, the lychee fruit borer (LFB) *Conopomorpha sinensis* Bradley (Lepidoptera: Gracilariidae), on the other hand is the most important lepidopteran borer affecting lychee and longan producing area in Southeast Asia. LFB larvae damages the fruit of lychee and longan causing up to 80% fruit decay rate [6].

Various control measures have been implemented to manage populations of these pests however they are either labor-intensive and costly (e.g., sleeving the cocoa pods with plastic bags), ineffective or viable (e.g., biological control methods such as the use of natural enemies and sex pheromones) [4] or has a negative impact on the environment (e.g., the use of chemical insecticides) [5, 7]. To reduce the amount of insecticides used, autocidal control tactics such as the sterile insect technique (SIT) could be integrated with existing control methods as part of an area-wide integrated pest management (AW-IPM) approach. However, determining the best approach for applying these strategies requires in-depth knowledge of the biology and ecology of the targeted pest populations, i.e., population size, reproductive strategies and dispersal. Information concerning the number of CPB populations present, and the degree of connectivity between them, is vital to constructing management and control policies for CPBs. For example, eradication may be possible in a series of small and discrete populations without a risk of reintroduction, whereas in geographically large populations, population reduction may be the best practicable approach [8, 9]. Such knowledge can be relatively easily assessed by the study of genetic variation of molecular markers between individuals, and within and between subsamples in space and time [10, 11].

Examples of molecular markers used for insect population studies are: mitochondrial DNA cytochrome oxidase I (mtDNA COI), coding nuclear elongation factor-1α (EF-1α) marker, randomly amplified polymorphic DNA (RAPD), restriction site associated DNA (RAD) and microsatellites [12, 13]. Microsatellite markers are short tandem DNA repeats of one to six nucleotides and are widely used for population genetic analysis, as they are abundant in most genomes, are highly polymorphic, Mendelian, co-dominant, and can be easily amplified by polymerase chain reaction (PCR) [14]. However, no microsatellites had been developed for any of the known *Conopomorpha* spp., and earlier attempts to analyze their population structure using COI and EF-1α remained inconclusive [15, 16].

In the present paper, we described the development and characterization of microsatellite markers of *C. cramerella* that can be used in further population genetics studies. We also tested these markers on a closely related species, the lychee fruit borer (LFB) *Conopomorpha sinensis*, a serious pest of lychee orchards [17, 18].

## Materials and methods

### Insect collection and DNA extraction

Adult CPB moths were collected from Malaysia (Mal), Indonesia (Indo), and the Philippines (Phil) (Table 1). In Malaysia and Indonesia CPB were hand-caught under infected cacao branches using a cylindrical plastic bag. In the Philippines, insects were collected from infected cacao pods or adults were trapped with pheromone traps. In Vietnam, LFB individuals were collected from lychee fruits. The samples were stored in 95% ethanol (and replaced with propylene glycol before shipment) and shipped to the Insect Pest Control Laboratory (IPCL) of the Joint FAO/IAEA Centre of Nuclear Techniques in Food and Agriculture, Seibersdorf, Austria. Upon arrival, the propylene glycol was replaced with absolute ethanol and stored at -20˚C.

Male and female individuals were distinguished by examining the ventrocaudal segments of the abdomen [19]. The total genomic DNA was extracted from one adult moth using Qiagen DNeasy® Blood & Tissue Kit (Qiagen Inc., Valencia, CA) according to the manufacturer's instructions. The quantity and quality of extracted DNA were assessed with a Synergy™ H1 microplate reader (BioTek, Winooski, Vermont, USA).

### Sequence analysis and microsatellite loci selection

The partial genome sequence of *C. cramerella* (Accession SJJU01000000, WGS SJJU01000001-SJJU01073142) available at the NCBI database was used as a reference for microsatellite primers selection. Msatcommander 1.08 [20] was used to search for di- and trinucleotide motifs. A total of 192 primer pairs were selected based on product size (180–300 nt) and the number of repeats ($\geq$ 11 repeats), synthesized, and tested for microsatellite amplification with PCR using the CPB extracted DNA (S1 Table).

The 192 synthesized primer pairs were screened on the DNA extracted from three individuals from Malaysia. PCR amplifications were performed by mixing 12.5 μL of Qiagen *Taq* PCR Master Mix Kit (Qiagen Inc., Valencia, CA), 10 μL of nuclease-free $H_2O$ (Qiagen Inc., Valencia, CA), 0.2 μM final concentration of each forward and reverse primer and 1.5 μL (4 ng) DNA. The PCR conditions were as follows: 94˚C for 2 min; 34 cycles at 94˚C for 15 s, 58˚C for 15 s, and 68˚C for 15 s; ending with a final extension for 5 min at 68˚C. The PCR amplification was checked on 2% agarose E-gel™ stained with ethidium bromide (Invitrogen, ThermoFisher Scientific, Waltham, MA). Of 192 primer pairs, 53 successfully amplified the expected microsatellite fragments (S2 Table). The 53 primer pairs were then screened on 30 individuals of *C.*

**Table 1. The locations, the geographical coordinates, and the number of adults of cacao pod borer (CPB) *Conopomorpha cramerella* and lychee fruit borer *Conopomorpha sinensi*s (LFB) samples used for microsatellite development.**

| Species | Country of origin | Sampling location | Latitude | Latitude | Collected moths | | |
| --- | --- | --- | --- | --- | --- | --- | --- |
| | | | | | Females | Males | Total |
| *C. cramerella* | Indonesia | Tarengge, Sulawesi | -2.51029 | 120.80766 | 12 | 12 | 24 |
| | Malaysia | Bagan Datuk | 3.97197 | 100.753445 | 12 | 12 | 24 |
| | Philippines | Kabacan, Cotabato | 7.14839 | 124.85229 | 12 | 12 | 24 |
| *C. sinensis* | Vietnam | Luc Ngan, Bac Giang province | 02363299 | 0665936 | 11 | 11 | 22 |

*cramerella* from three locations (10 individuals per location), and 12 primer pairs were selected based on consistent amplification across all individuals and expected fragment size (Table 2).

## Genotyping of microsatellite loci

A total of 72 individuals of CPB from three locations (**Table 1**) were used for microsatellite characterization. Microsatellite amplifications were performed by using a 12.5 μL Qiagen *Taq* PCR Master Mix Kit (Qiagen Inc., Valencia, CA), mixed with 9.6 μL nuclease-free $H_2O$ (Qiagen Inc., Valencia, CA), 0.016 μM forward primer with M13-adapter (5'–CACGACGTTGTAA AACGAC–3'), 0.2 μM reverse primer, 0.2 μM M13 adapter labeled with fluorescent dye (6-FAM for Cpb14, Cpb84 and Cpb136, HEX for Cpb55, Cpb122 and Cpb160, ATTO 550 for Cpb54, Cpb112 and Cpb139 or ATTO 565 for Cpb62, Cpb135 and Cpb190) and 1.5 μL diluted DNA ($\sim$ 4 ng/uL). We used the Platinum™ II Hot Start PCR 2X Master Mix (Thermo Fisher Scientific, Waltham, MA) instead of Qiagen *Taq* PCR Master Mix for Viet samples due to its higher sensitivity. Amplifications were done using 34 cycles and annealing temperatures given in **Table 2**. PCR amplicon products were resolved in 4% agarose E-gel™ stained with ethidium bromide (Invitrogen, Thermo Fisher Scientific, Waltham, MA) and genotyped using ABI 3500XL Genetic Analyzer (Applied Biosystems™, Foster City, CA) with a GeneScan™ 600 LIZ™ dye size standard (Thermo Fisher Scientific, Waltham, MA). Allele calling was performed

**Table 2. Primer details from Msatcommander and optimized annealing temperature of 12 polymorphic microsatellites of *Conopomorpha cramerella*.**

| Microsatellite | Primer sequence (5' - 3') | Repeat motif and number | Product size, bp | Annealing temp,˚C |
|---|---|---|---|---|
| Cpb14[*] | F: CTGTAGAGTGCGGAGTGTCG | $(CCG)_{19}$ | 264 | 60 |
| | R: TTTGCTCGCTGTTAGGTCGG | | | |
| Cpb54 | F: GGTAAGAGTTGCGGAATGGC | $(AC)_{14}$ | 264 | 55 |
| | R: TCGCGGGAATAAGGGCAC | | | |
| Cpb55[*] | F: TGACTAAGCACCCTCTCACG | $(CCG)_{14}$ | 273 | 55 |
| | R: ATAGCCCAGAACCACCCTTC | | | |
| Cpb62 | F: CAGAACACAGATCGTTGCCC | $(AC)_{13}$ | 213 | 61 |
| | R: CATGGCGATGAAAGTGATTGC | | | |
| Cpb84 | F: CACACAGCTAAGCGAACACC | $(AC)_{12}$ | 189 | 61 |
| | R: CTTCAACGCTCATCACCTGC | | | |
| Cpb112 | F: TCGGCCGTCTCGAGATATTC | $(AC)_{12}$ | 288 | 60 |
| | R: CTCAGAAATGGTGACCCGTG | | | |
| Cpb122 | F: AAGCAAATTGTCACCGACCC | $(AC)_{11}$ | 189 | 55 |
| | R: GCCGGGCACTTTACTTGATG | | | |
| Cpb135* | F: TGTAATCGGCCCACTTCCTC | $(CCG)_{11}$ | 214 | 56 |
| | R: TCGGAGATGGATCGTGTCTG | | | |
| Cpb136[#] | F: TGTAATCGGCCCACTTCCTC | $(AC)_{11}$ | 218 | 55 |
| | R: TGCGACGTTGTTACACTTCG | | | |
| Cpb139 | F: GTCATTTCACCGACGACTATGG | $(CCG)_{11}$ | 228 | 55 |
| | R: AACCCACCGATTCCAGAGAG | | | |
| Cpb160 | F: GTTGACGTGACCCATATGCG | $(AC)_{11}$ | 257 | 55 |
| | R: TCGGATAGCGTTTCGAGTGG | | | |
| Cpb190 | F: CTGTTGTTGAGCCGTTCCTG | $(AC)_{11}$ | 284 | 60 |
| | R: CTCACACATCCTGGCGAATG | | | |

[*]: Loci excluded from quality testing due to amplification failures.

[#]: Msatcommander prediction indicated these loci as tri-nucleotides however genmapper alleles indicated di-nucleotides.

using GeneMapper 4.1 software (Applied Biosystems™, Foster City, CA) to obtain a co-dominant allele matrix from the raw data. The genetic data were formatted for Create 1.37 [21] to convert the datasets into the required formats according to needs.

In Lepidoptera, sex determinism consists of female heterogamety, with a majority of species displaying the WZ/ZZ (female/male) mechanism [22]. Following this, any polymorphic locus located in one of the two sex chromosomes should display strong discrepancies of heterozygosity between females and males. In absence of recombination for the W chromosome, we also expect an accumulation of divergent mutations at any locus located on a sex chromosome between males and females. To check for chromosomal location, we have measured the heterozygote deficit with Weir and Cockerham's (Weir and Cockerham 1984) unbiased estimator of Wright's [23] $F_{IS}$ in males and females. To limit the number of tests, we compared males and females for the three loci (Cpb14, Cpb112 and Cpb122) displaying the biggest $F_{IS}$ differences with a Wilcoxon signed rank test for paired data, the pairing unit being the allele. These tests were undertaken with R-commander (Fox, 2005, 2007) for R (R-core Team). For genetic divergence, we used Weir and Cockerham's unbiased estimator of Wright's $F_{ST}$ measured between females and males within each country and for each locus, and tested the significance of this divergence with the $G$-based randomization test (Goudet et al 1996), after 10,000 randomizations of individuals between subsamples. These computations were undertaken with Fstat 2.9.4 [24]. We averaged $F_{ST}$s across countries and combined the corresponding $p$-values with the generalized binomial procedure with MultiTest [25]. False discovery rate across the 11 loci was then taken into account with the Benjamini and Hochberg procedure [26] with the command "p.adjust" of R.

## Detection of outlier individuals in the different subsamples

To check if we have outliers in CPB and LFB samples, we first used the Bayesian clustering algorithm of STRUCTURE 2.3.4 [27]. For this analysis data were converted with Convert [28]. We used a burn-in period of 5,000 and 50,000 Markov chain Monte Carlo (MCMC) iterations, asked for a number of clusters $K$ between 1 and 10, with 10 iterations. The number of genetic clusters was determined following the Delta-K method [29] using Structure Harvester [30]. Lastly, to visualize the genetic relationships between individuals, a genetic distance matrix was computed using Cavalli-Sforza and Edwards' chord distance ($D_{CSE}$) [31], corrected for null alleles with the INA procedure with FreeNA [32]. This matrix was used to build a neighbor-joining tree (NJTree) [33] with Mega 7 software [34] as recommended by Takezaki and Nei [35]. This version of MEGA indeed allow to import the resulting tree into an object that is editable with a drawing/presentation program (Impress, PowerPoint, etc.), while more recent versions only create an image that cannot be modified.

## Quality assessment of microsatellite loci

The following analyses were undertaken on the two species separately. The $G$-based tests for linkage disequilibrium (LD) between each pair of loci [36] were conducted with 10000 randomizations, over all subsamples. This produced as many non-independent tests as there are locus pairs. The series of $p$-values was thus adjusted using the Benjamin and Yekutieli (BY) false discovery rate (FDR) procedure [37] in RStudio 2021.09.2 [38]. Proportions of significant tests in the different sites were compared with a Fisher exact test under rcmdr.

The deviation from panmictic expectations of genotypic frequencies within subsamples (Wright's $F_{IS}$), and subdivision (Wright's $F_{ST}$) were estimated using Weir and Cockerham's unbiased estimators [39]. The significance of deviations from panmixia and of subdivision were tested with 10000 permutations of alleles between individuals within subsamples and of

individuals between subsamples, respectively. For $F_{IS}$, tests are one-sided (heterozygote deficits) by default. We computed two-sided $p$-values by doubling the $p$-value in case of positive $F_{IS}$, or by doubling (1-$p$-value) otherwise. For subdivision, tests were kept one-sided. The confidence intervals were computed after 5,000 bootstraps over loci. Wright's $F_{IT}$ was also estimated for short alleles dominance (SAD) analysis (see below).

In case of significant heterozygote deficits, we used the approaches described in several papers [40–43]. For null alleles, the ratio of standard errors of $F_{IS}$ over $F_{ST}$ ($r_{FIS/FST}$) was computed, a value above 2 being a signature of probable null alleles. We also measured the correlation between $F_{IS}$ and $F_{ST}$, and between $F_{IS}$ and missing data ($N_{Blanks}$: putative null homozygotes), which are expected to be positive in case of null alleles. These correlations were measured and tested with one-sided Spearman's rank correlation test with rcmdr. We also computed the slope, intercept and determination coefficient of the regression $F_{IS} \sim N_{Blanks}$. Expected frequencies of null alleles followed the expectation-maximization step (EM) algorithm with FreeNA and were compared to observed ones. Stuttering signatures were checked with the recently developed methods of De Meeûs and Noûs [43] and corrected by pooling alleles with one to two base differences, following rules defined in [41]. The presence of SAD was checked using a one-sided (negative correlation) Spearman's rank correlation test between $F_{IT}$ and allele size [42]. In case of doubt, we also undertook the regression approach $F_{IS} \sim$ Allele size weighted by $p_i(1$-$p_i)$, where $p_i$ is the frequency of allele of size $i$ [44]. The goodness of fit of expected null homozygotes and observed missing data was tested with a one-sided exact binomial test with RStudio 2021.09.2 [38] (command "binom.test" with the alternative = "less"). In all cases, independent test series were adjusted with the Benjamini and Hochberg (BH) procedure [26] with RStudio (**S1 File**).

## Population genetics structure of *Conopomorpha* samples

These analyses (subdivision, population size, number of immigrants, immigration rates and dispersal distance) were undertaken for each species separately, and after cleaning datasets from outlier loci or individuals. Because of the presence of null alleles, we used FreeNA to measure subdivision with the excluding null alleles (ENA) correction for null alleles, after recoding missing data as homozygotes for the null allele (999999) when relevant [32]. We computed 95% confidence intervals (CI) with 5000 bootstraps over loci. We corrected this estimate for excess of polymorphism using Meirmans' method with RecodeData [45] to compute the maximum possible divergence $F_{ST\_max}$ with Fstat. The standardized estimate was then computed as $F_{ST\_FreeNA}/F_{ST\_max}$. For the record, we did not use Meirmans and Hedrick's $G_{ST}$" [46], because it cannot correct for the presence of null alleles. Significance was assessed with the $G$-based permutation test with Fstat and false discovery rate correction was undertaken with Benjamini and Yekutieli (Benjamini and Yekutieli) (BY) correction with the command "p.adjust" of R.

Effective population sizes ($N_e$) were estimated with five different methods. The first method ($N_eF_{IS}$) used a recent $F_{IS}$ based method where $N_e$ = -1/(2$F_{IS}$)-$F_{IS}$/[2(1+$F_{IS}$)] [43]. Because some loci with null alleles and heterozygote deficits could not be used, we averaged the values obtained across loci displaying an heterozygote excess. The second method (LD$N_e$) was the LD based method [47], adjusted for missing data [48], where only alleles with frequency above 0.05 were used and random mating assumed. The third method (CoA$N_e$) corresponded to the co-ancestry method [49]. These two methods were computed with NeEstimator 2.1 [50]. The fourth method (I$N_e$) was computed with ESTIM 1.2 updated from [51] which uses the one and two locus identity probabilities based effective population size [52, 53]. The last one is the sibship frequency based effective population size (S$N_e$) [54], assuming polygamy and inbreeding, computed with Colony 2.0.6.8 [55]. All methods but I$N_e$ assume isolated subsamples,

while INe assumes an Island model of migration. Non-isolation should tend to generate under-estimates for small subpopulations with rather small immigration rates, at least for $N_eF_{IS}$, and $LDN_e$, while over-estimates could be expected with $CoAN_e$ and $SN_e$. For $IN_e$, alternative population structure as isolation by distance should produce over-estimates as well. Nevertheless, these biases are not expected to change the order of magnitude of estimates. For each method, we computed the average and minimum and maximum values. We then computed the unweighted grand average across methods of these three quantities. This hopefully ensured a range of values that contained real values.

The effective number of immigrants exchanged between subsamples was then calculated: globally, assuming an infinite island model, with $N_em = (1-F_{ST\_FreeNA}')/(4F_{ST\_FreeNA}'))$ (e.g. De Meeûs 2021 [56], page 52, equation 24, assuming $m >> u$). Geographic distances between all pairs of subsamples ($D_{geo}$) were computed from the GPS coordinates in decimal degrees using the geosphere package [57] of R. Finally, we estimated the immigration rates ($m = N_em/N_e$) and dispersal distances ($\delta = m$ x $D_{geo}$) where $D_{geo}$ is the geographic distance, either between two subsamples or the average between all subsamples.

**Inclusivity in global research.** Additional information regarding the ethical, cultural, and scientific considerations specific to inclusivity in global research is included in the S2 File.

## Results

### Markers development and validation

The Msatcommander search for the di- and tri nucleotides motifs in the genome of the CPB showed a total of 37,532 motifs and 35,076 pairs of primers. Out of those, 6,346 primer pairs were found in duplicates in the genome and therefore excluded from the analysis. Combining the unique primer pairs with the motifs using vlookup in Excel produced 28,730 primer pairs of which 9,968 primer pairs amplified di-nucleotides repeats and 18,762 amplified trinucleotide repeats. Of these, 7,155 primer pairs produced PCR products ranging between 180 and 300 nucleotides of which 203 primers pairs amplified motifs with $\geq$ 11 repeats. A total of 192 primer pairs were then synthesized and tested by PCR with a pooled DNA extract from three CPB adults from Malaysia (S1 Table).

Of the 192 primer pairs, 53 primers amplified a PCR product with the expected size (S2 Table). Testing these 53 primer pairs on the DNA of LFB collected from Vietnam indicated that only 29 primer pairs amplified the expected PCR product (S2 Table). Out of the 29 primer pairs that amplified the expected PCR product for CPB and LFB, 12 primer pairs showed heterogeneity between tested individuals and were therefore used for the initial population genetics study. One of these loci (Cpb136) displayed more than 50% amplification failures and was also disregarded.

Data obtained with the remaining 11 loci are presented in S3 Table. With the 11 remaining loci, none of the $F_{IS}$ comparisons between females and males for CPB and LFB showed a significant value (smallest $p$-value > 0.09094). No locus provided a significant genetic divergence between genders (minimum $p$-value > 0.1661) (S4 and S5 Tables). We thus assumed all these 11 loci to be autosomal.

### Detection of outlier individuals in the different subsamples

Most individuals were assigned with very high probabilities (>0.99) to either LFB in Vietnam or CPB in the other sampling locations (Fig 1). However, two individuals did not follow this rule, i.e., individual 77 from Vietnam clearly belonged to CPB, and individual 53 from Indonesia could represent a hybrid between the two species.

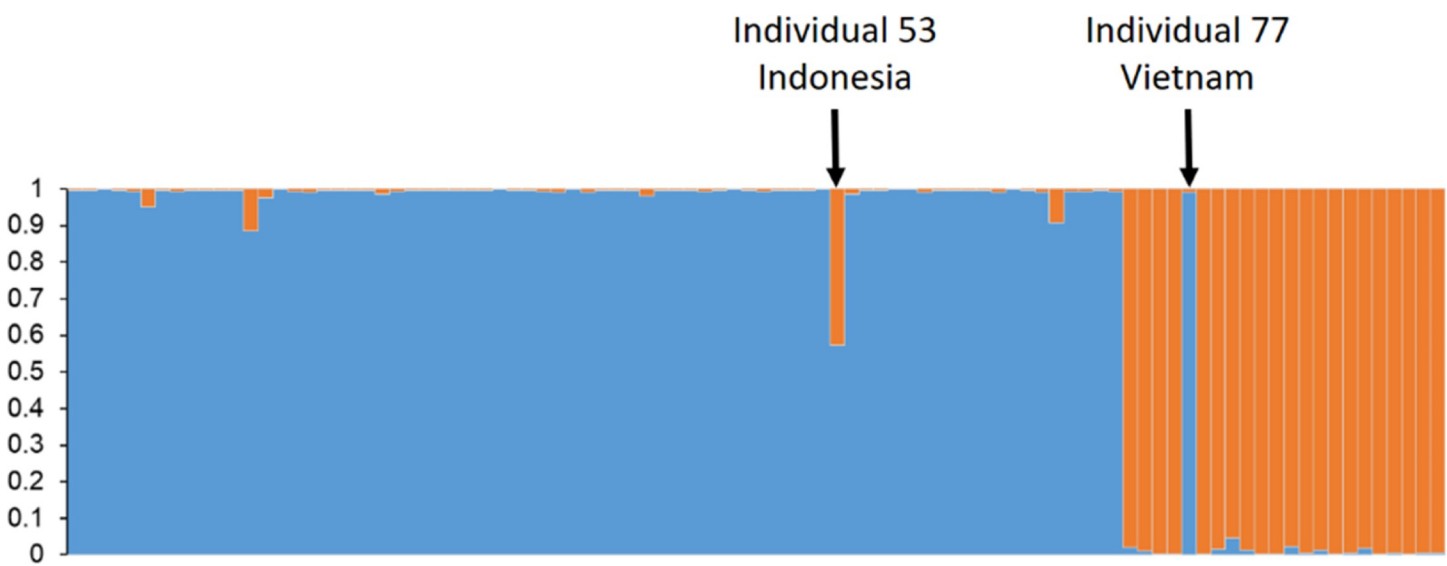

**Fig 1. Assignment probabilities of each individual of *Conopomorpha cramerella* and *C. sinensis* to the two optimal clusters obtained with STRUCTURE and STRUCTURE Harvester.** Results are presented for the optimal *K* = 2.

The NJTree analysis did not confirm the hybrid status of individual 53, but confirmed that individual 77, sampled from lychee fruits from Vietnam belongs to *C. cramarella* (Fig 2), obviously represented a misplaced individual. Whatever the cause (immigrant from a cacao plantation, mislabelling or manipulation error), we removed individual 77 from further investigations. Only one locus appeared diagnostic between the two species, locus Cpb62, with alleles 191 and 211 that are only found in LFB, while alleles 185, 205, 217 to 233 (with one repeat increment), 237 and 257, were only found in CPB.

## Quality assessment of microsatellite loci for *C. cramerella*

Proportions of significant LD tests did not significantly vary across subsamples ($p$-value = 0.1538). Over all subsamples, eight locus pairs displayed a significant LD (13%). Three locus pairs stayed significant after BY correction: Cpb55 x Cpb54, Cpb135 x Cpb160 and Cpb139 x Cpb160. The loci involved in these three significant pairs will thus need special attention.

We observed a highly significant and highly variable $F_{IS}$ ($F_{IS}$ = 0.285, $p$-value > 0.0002, 95% CI = [-0.021, 0.261]) (Fig 3). The ratio $r_{FIS/FST}$ = 8 suggested that null alleles may have explained a part of this deficit. The correlation between $F_{IS}$ and $F_{ST}$ was positive but not significant ($\rho$ = 0.4727, $p$-value = 0.0728), and the correlation between $F_{IS}$ and $N_{Blanks}$ was negative. However, there was quite a good agreement between expected and observed missing data, since only locus Cpb135 displayed a significant deviation ($p$-value = 0.02665), which did not stay significant after BH correction. Obviously, null alleles cannot explain all heterozygote deficits. Only a single SAD correlation test was significant, for locus Cpb160 ($p$-value = 0.04575), which was not confirmed by the regression test (one sided $p$-value = 0.2877). Finally, no loci displayed a significant stuttering (all $p$-values > 0.13).

Given that some loci display negative values, as expected in dioecious species [58–60], and other loci heterozygote deficits, locus specific causes, unlike a Wahlund effect, probably better explain our data. This was confirmed by the substantially variable and highly significant $F_{ST}$ = 0.043 in 95% CI = [0.027, 0.06], with several loci displaying very small or very high values that

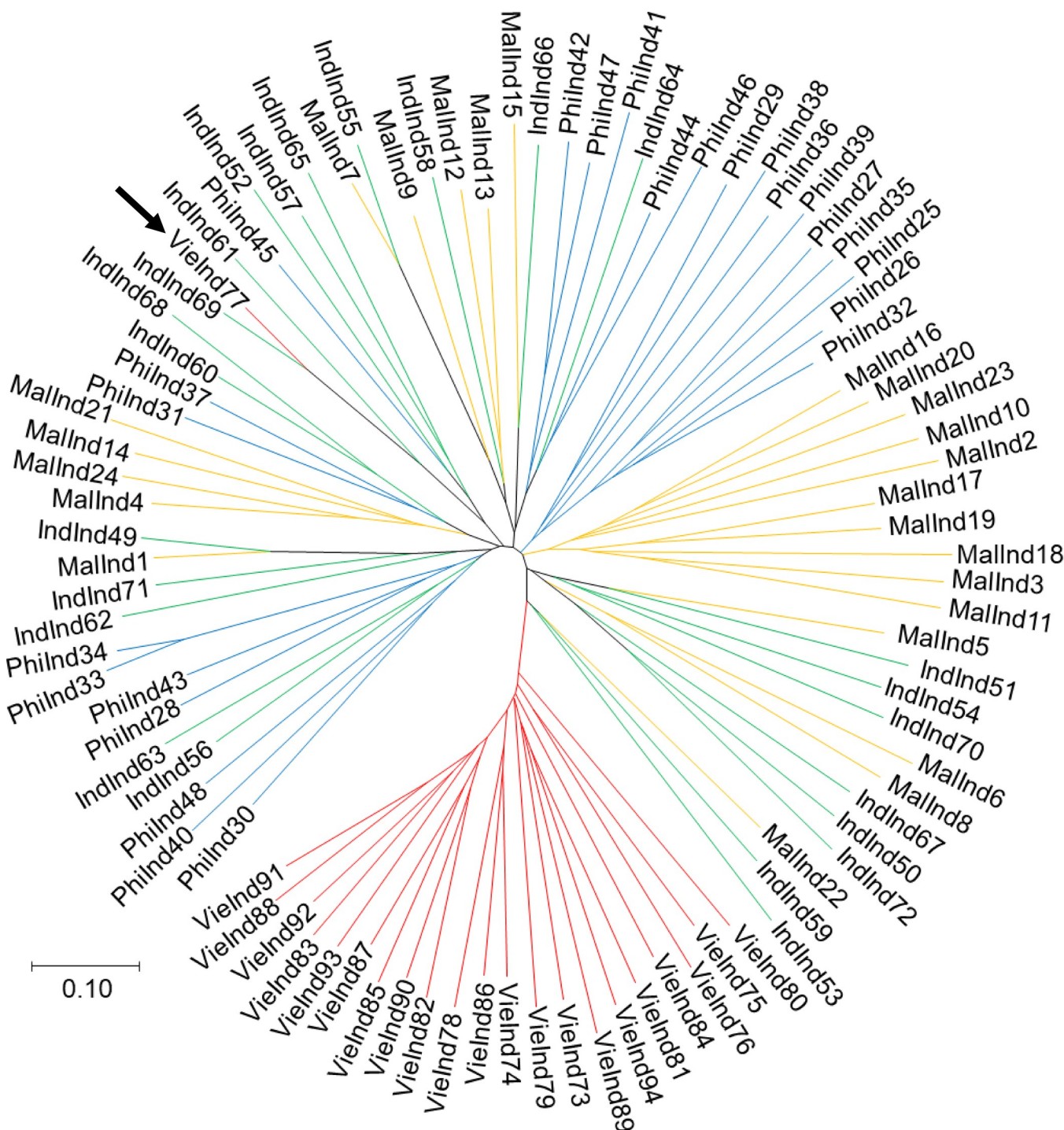

**Fig 2. Neighbor-joining tree based on Cavalli-Sforza and Edward's chord distance between individuals of the two species of *Conopomorpha*, and corrected for null alleles.** Individual names are composed of the three first letters of the site of origin (Ind = Indonesia, Mal = Malaysia, Phi = Philippines and Vie = Vietnam), followed by the individual number. Sites are also indicated by different colors of the branches (Indonesia: green; Malaysia: Yellow; Philippines: Blue; and Vietnam: red).

ranged outside the 95% CI: loci Cpb14, Cpb55, Cpb62, Cpb84, Cpb135, Cpb139 and Cpb190 (**Fig 4**). Moreover, three locus pairs were found in significant LD and display very extreme $F_{IS}$

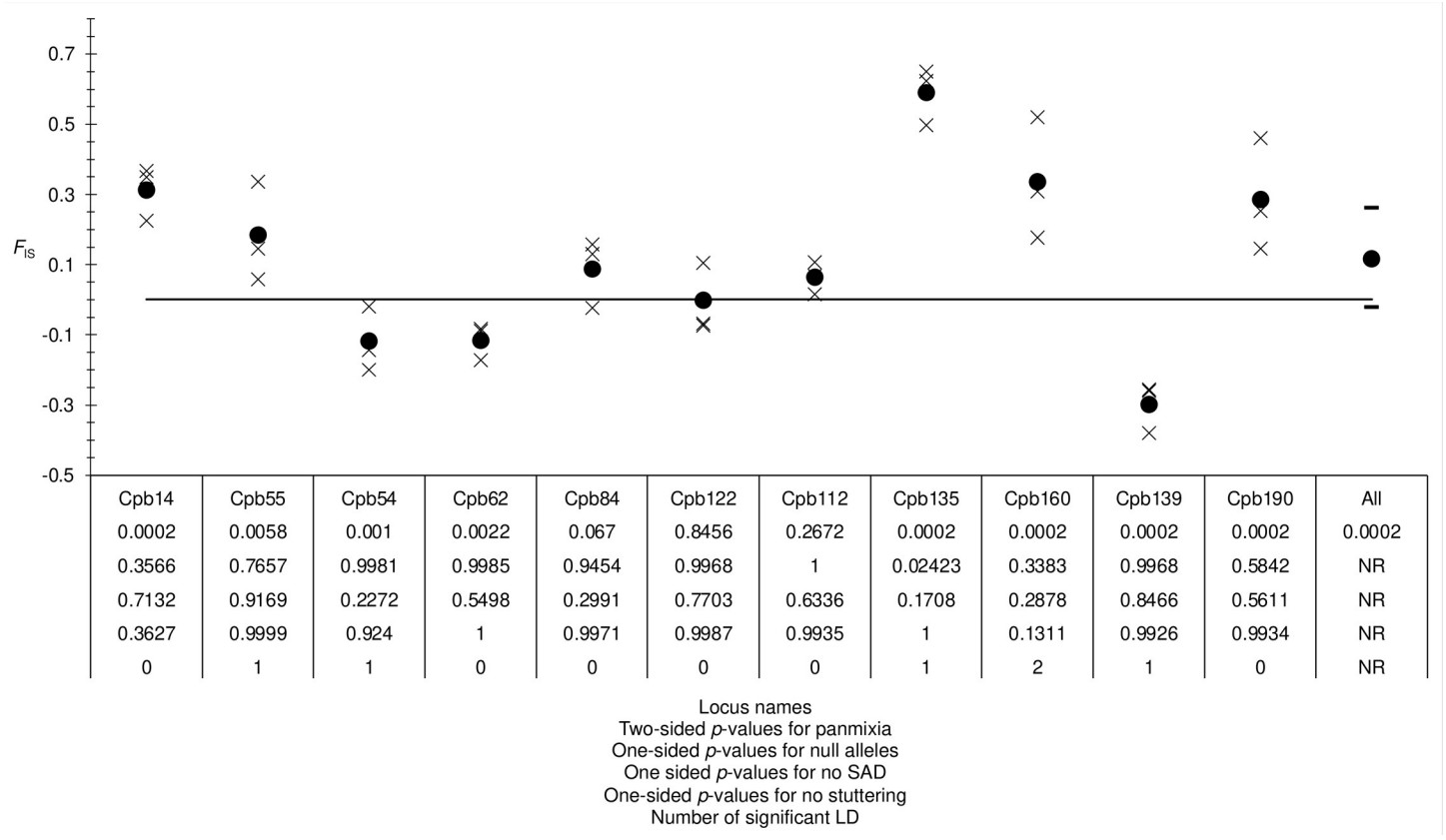

**Fig 3. Variation of $F_{IS}$ across the 11 loci for *Conopomorpha cramerella* subsamples, and over all loci (All).** The 95% CI of bootstraps over loci is also represented (dashes), and the *p*-values for no subdivision can be seen under locus names. Under these *p*-values, result of tests for null alleles, SAD, stuttering and LD were also inserted.

(**Fig 3**). Locus Cpb139 was an outlier for both $F_{IS}$ and $F_{ST}$ and was involved in a significant LD pair. Locus Cpb14 displayed a negative $F_{ST}$ and was also an outlier for $F_{IS}$. Cpb55 was an outlier for $F_{ST}$ and in significant LD, and Cpb135 was an outlier for $F_{ST}$, $F_{IS}$ and for LD. All these complementary observations suggest that these loci are under selection of some sort (homogenizing or disruptive, depending on the locus) and we decided to eliminate these loci in order to limit as much as possible all biases of this kind.

The selected seven CPB microsatellite loci (Cbp54, Cbp62, Cbp84, Cbp112, Cbp122, Cbp160 and Cbp190) were characterized and the results indicated that the number of alleles ranged from 12 (Cpb190) to 19 (Cpb84) with an average of 16 alleles per locus. The total genetic diversity (Ht) ranged from 0.85 (Cpb190) to 0.896 (Cpb84) with an average of 0.869 (Table 3). Wright's *F*-statistics were $F_{IS}$ = 0.072, *p*-value = 0.0002, 95% CI = [-0.042, 0.197])), and $F_{ST}$ = 0.044, *p*-value = 0.0001, 95% CI = [0.026, 0.067]) (**S1 Fig**). The correlation between the $F_{IS}$ values and the number of missing genotypes ($N_{Blanks}$) displayed a negative slope (Fig 5). Cpb62 and Cpb122 had a negative $F_{IS}$ with 2 and 9 missing data respectively. Obviously, these missing data are not null homozygotes since they should display 0 null homozygotes, hence 0 missing data due to null alleles. If we ignore these loci from the regression, the slope becomes positive and the correlation between $F_{IS}$ $N_{Blanks}$ is now significant ($\rho$ = 0.8660, *p*-value = 0.0288) (**S2 Fig**). Moreover, we noticed that the averages of null frequency estimates (**S6 Table**) are far from the 0.20 threshold described in Séré et al. [61], and that null alleles explained 80% of $F_{IS}$ variation, with an intercept ($F_{IS}$ with no null alleles) $F_{IS\_0}$ = -0.03.

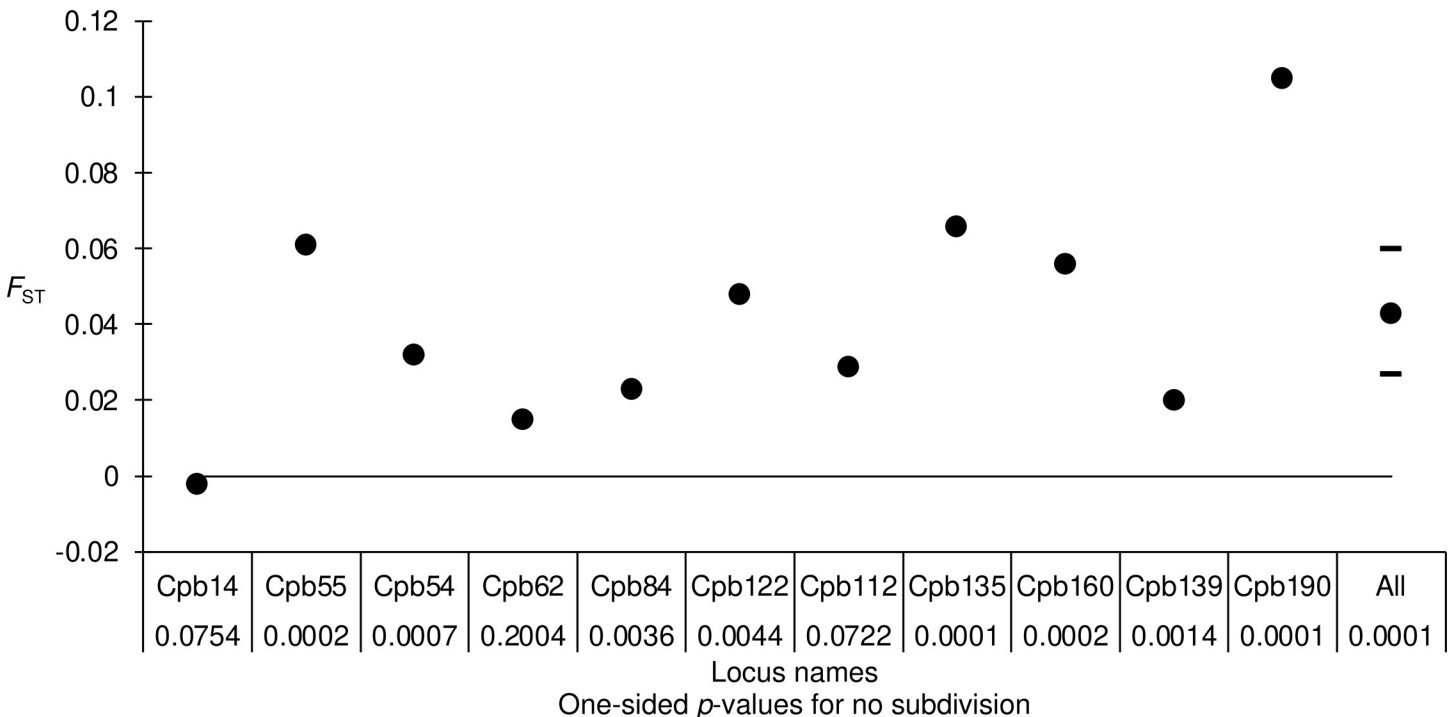

**Fig 4. Variation of $F_{ST}$ across the 11 loci for *Conopomorpha cramerella* subsamples, and over all loci (All).** The 95% CI of bootstraps over loci is also represented (dashes), and the *p*-values for no subdivision can be seen under locus names.

## Population structure of *C. cramerella* at the seven selected loci

For this analysis, we kept loci Cpb54, Cpb62, Cpb84, Cpb122, Cpb112, Cpb160 and Cpb190, with 0, 2, 0, 9, 0, 1, and 1 missing data respectively, and recoded missing data as homozygous for the null allele (999999) for only loci Cpb160 and Cpb190. Globally, $F_{ST}$ = 0.044 in 95% CI = [0.026, 0.067] was highly significant (*p*-value < 0.0001) and its variation mainly explained by null alleles ($R^2$ = 0.67 for the regression of $F_{ST}$ and $N_{Blanks}$) (S2 Fig). After correction for null alleles and excess of polymorphism, we obtained $F_{ST\_FreeNA}$' = 0.2629 in 95% CI = [0.1681, 0.3768], which translated in $N_e m$ = 0.7 in 95%, CI = [0.4, 1.2] individuals per generation. Given that this pest has a one month generation time all year long [62], and given the large distances between the subsamples (between 1159 and 2693 km), this potentially represents a

**Table 3. Characterization of the seven (7) polymorphic microsatellite loci of cacao pod borer *Conopomorpha cramerella*.**

| Locus | Repeat motif and number | % Amplification | Size range (bp)* | $N_A$ | $H_o$ | $H_s$ | $H_t$ |
|---|---|---|---|---|---|---|---|
| Cpb54 | $(AC)_{14}$ | 100 | 131–281 | 18 | 0.972 | 0.869 | 0.889 |
| Cpb62 | $(AC)_{13}$ | 97.22 | 185–257 | 13 | 0.957 | 0.858 | 0.867 |
| Cpb84 | $(AC)_{12}$ | 100 | 197–245 | 19 | 0.806 | 0.882 | 0.896 |
| Cpb122 | $(AC)_{11}$ | 87.5 | 184–226 | 18 | 0.854 | 0.846 | 0.877 |
| Cpb112 | $(AC)_{12}$ | 100 | 184–310 | 16 | 0.778 | 0.83 | 0.847 |
| Cpb160 | $(AC)_{11}$ | 98.61 | 264–302 | 13 | 0.547 | 0.826 | 0.859 |
| Cpb190 | $(AC)_{11}$ | 98.61 | 281–307 | 12 | 0.565 | 0.789 | 0.85 |

$N_A$ = number of alleles, $H_o$ = observed heterozygosity, $H_s$ = expected heterozygosity within samples, $H_t$ = expected heterozygosity between samples
* Allele size includes the M13 adapter tail (19 bp).

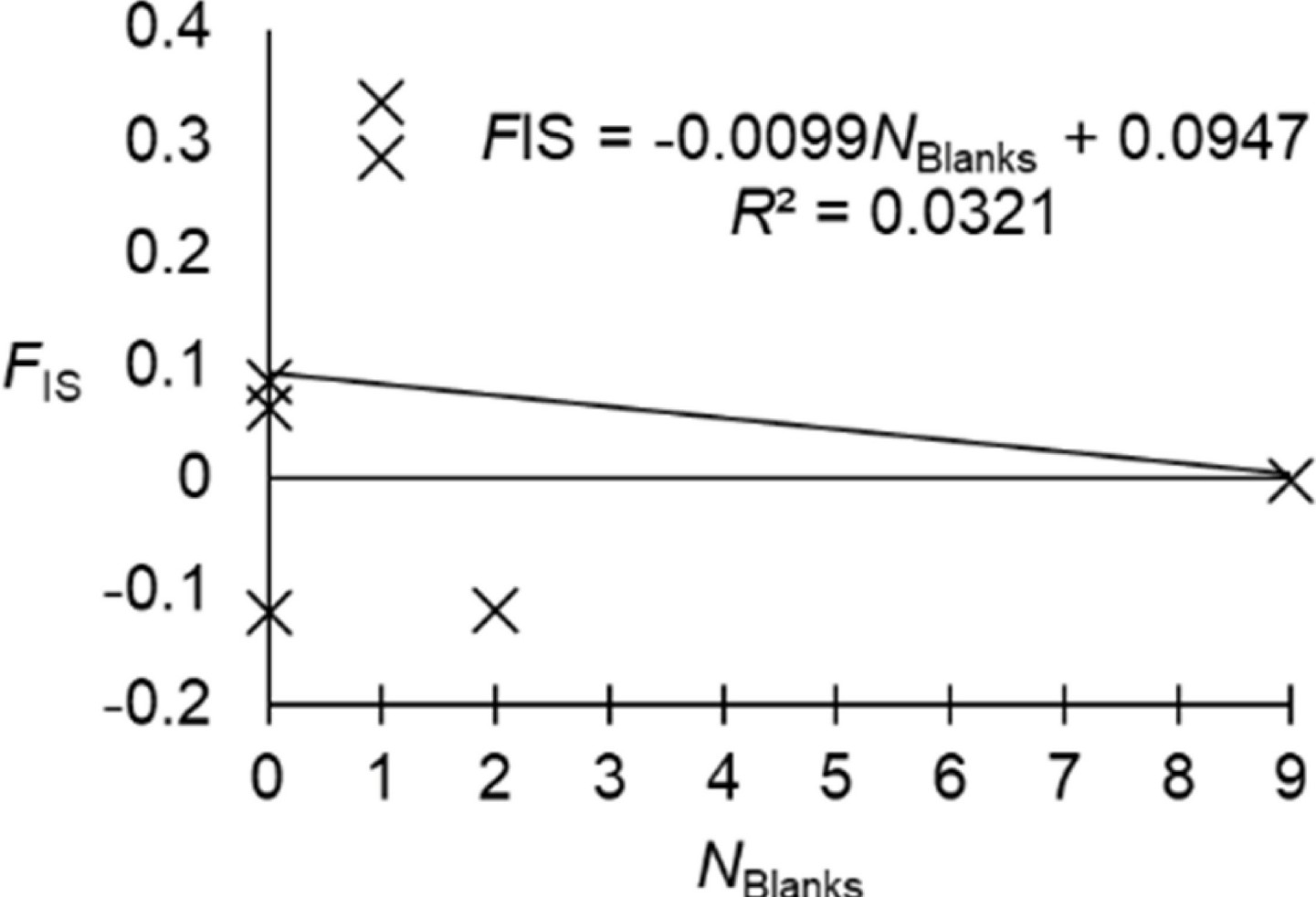

**Fig 5. Regression between $F_{IS}$ and number of missing genotypes ($N_{Blanks}$) for the selected seven loci of *Conopomorpha cramerella* after exclusion of putatively loci under selection.**

substantial exchange of 8 in 95% CI = [1, 14] individuals per year between the three countries. Effective population sizes were highly variable across methods and subsamples: average $N_e$ = 2128 in minimax = [45, 4228] (Fig 6). This translated into similar dispersal estimates assuming an Island model, with average distances between subsamples, or computed from paired sub-samples. On average $\delta \approx 1$ km per generation, with an important window of possible values: i.e. between 100 m and 60 km per generation (Fig 7).

## Quality assessment of microsatellite loci for *C. sinensis*

This analysis was carried out with the data from Vietnam, after exclusion of individual 77, which appeared to belong to CPB (see above). No significant LD was detected. We observed a highly significant heterozygote deficit, which was quite variable across loci ($F_{IS}$ = 0.124, *p*-value = 0.0001, 95% CI = [0.017, 0.242]) (Fig 8). No $F_{ST}$ or $F_{IT}$ were available (only a single sub-sample). SAD tests were undertaken with the weighted regression of $F_{IS}$ only. No evidence of SAD or stuttering could be found, and null alleles explained the data rather well. More missing data were observed as compared to those expected if null alleles explained all heterozygote deficits (all *p*-values > 0.7), and the correlation between $F_{IS}$ and missing data was significant ($\rho$ =

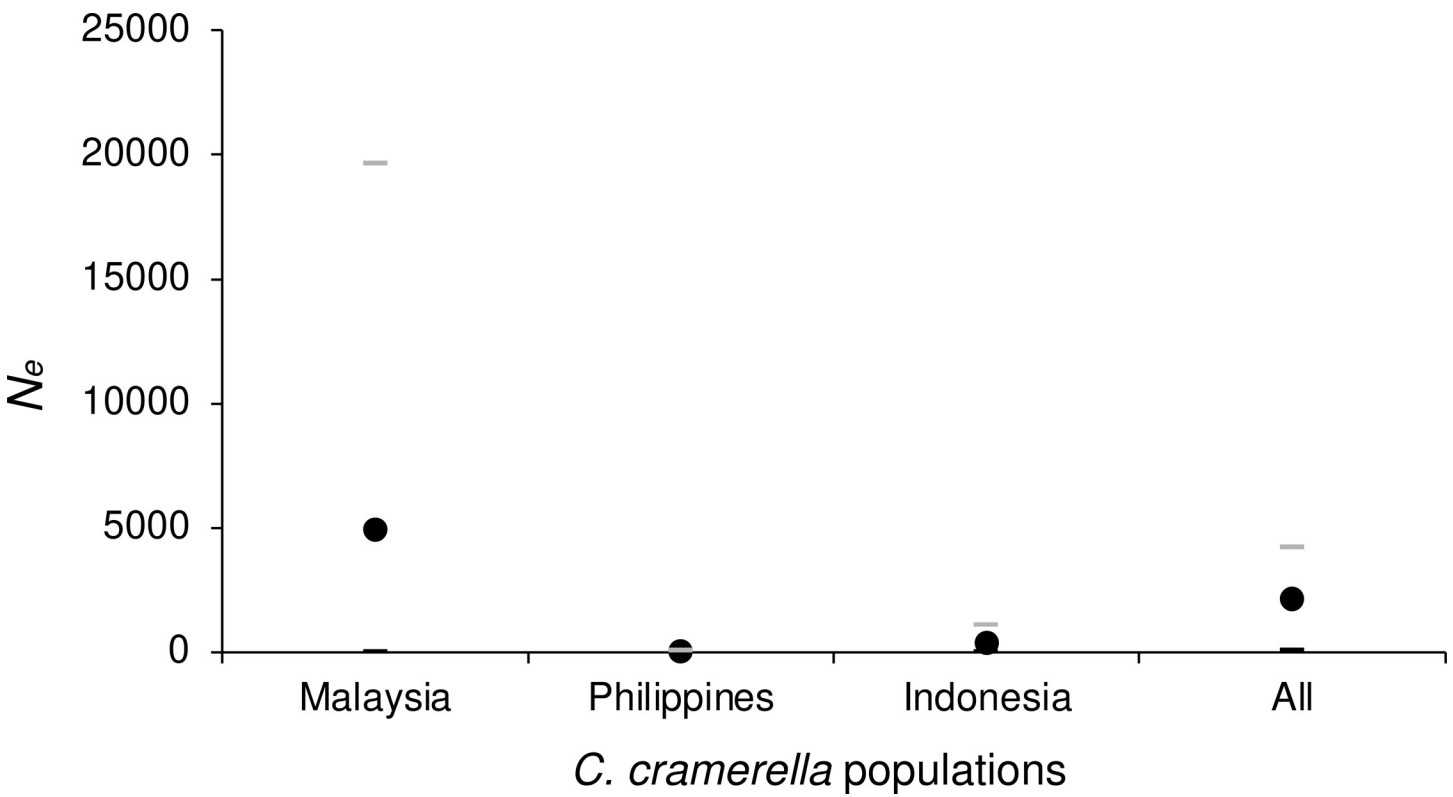

**Fig 6. Effective population sizes of the cocoa pod borer *Conopomorpha cramerella* populations using different methods.** [•]: Average population size, [–]: Upper and lower limits.

0.5861, *p*-value = 0.0290), and the corresponding regression model explained 48% of $F_{IS}$ variation (S3 Fig). On the other hand, using null allele frequency estimates, the average was also far from the threshold and null alleles explained 75% of $F_{IS}$ variation with $F_{IS\_0}$ = -0.031 (S6 Table). It can also be noted that locus Cpb62 was almost monomorphic ($H_S$ = 0.048).

### Population genetics structure of *C. sinensis* on the eleven loci

With a single subsample (Vietnam), we could only estimate effective population size. To obtain minimum and maximum values with $LDN_e$, $CoAN_e$, $IN_e$ and $SN_e$, we kept the 95% CI values

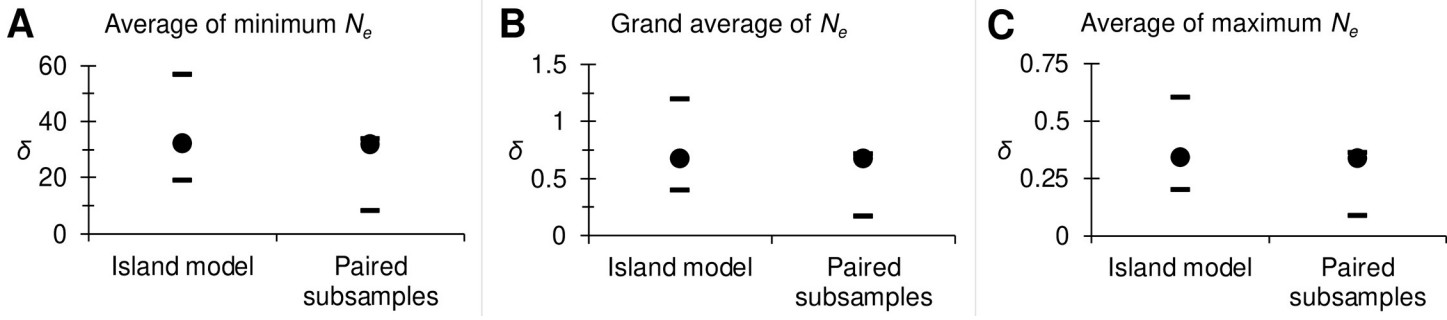

**Fig 7.** Dispersal estimates (δ) assuming an Island model and paired subsamples based on different population size ($N_e$) estimates: (A) minimum, (B) grand average and (C) maximum).

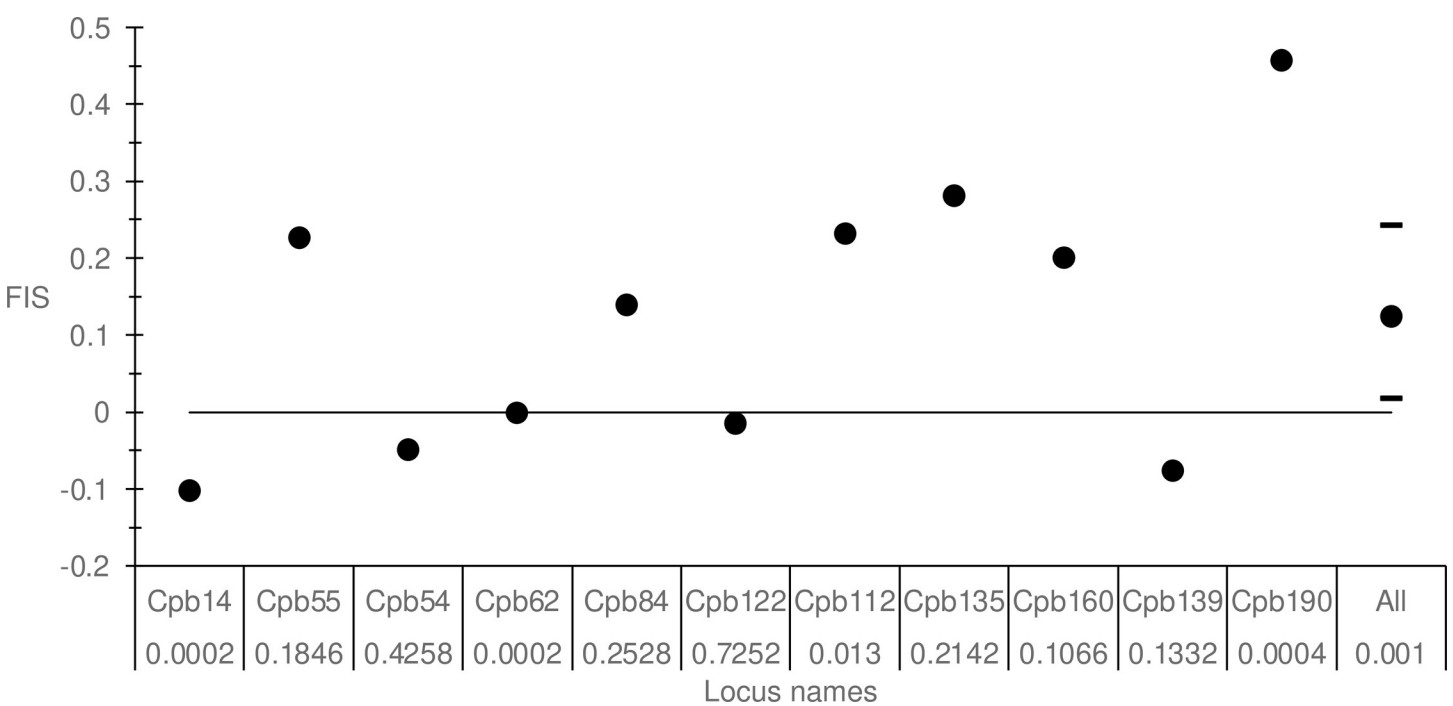

**Fig 8. Quality assessment of the cocoa pod borer *Conopomorpha cramerella* microsatellite loci on lychee fruit borer *Conopomorpha sinensi*s samples.** [•]: $F_{IS}$ per locus, [–]: Upper and lower limits.

provided by the output of the corresponding software. We observed on average relatively smaller values, with $N_e \approx 50$ in minimax = [16, 191], but in view that the overlap with CPB populations was substantial, nothing conclusive could be extrapolated.

## Discussion

The goal of this study was to develop new microsatellite markers that could effectively be used to explore the genetic differences between different populations of the CPB and test the selected microsatellites on a closely related species, the LFB. To the best of our knowledge, there were no microsatellite markers available for these insects, and therefore, no available information on the population's genetic structure of these pests. The availability of these microsatellite markers will allow analyzing the population genetics of this species and therefore assess the level of gene flow between infested countries in Southeastern Asia. This information would be of paramount importance for trading barriers and for the development of AW-IPM strategies that could potentially include the SIT to manage various populations of this pest.

The availability of the partial genome sequence of the CPB enabled us to search for potential microsatellite markers. The search for microsatellite markers indicated that more tri-nucleotides than di-nucleotides markers were found, which is contradictory to the theory that the microsatellite abundance decreases with the increase of motif repeat number and repeat length [63]. This might be due to the incomplete genome of the CPB that was used for the analysis, and the sequence that is missing might have contained more di-nucleotides sequences. On the other hand, our result agrees with the data of Liu et al. [64] who also obtained more frequent tri-nucleotides than di-nucleotides repeats in the hoverfly, *Eupeodes corollae*.

Marker development allowed to pre-select 11 loci that worked well on the two species, CPB and LFB. Following the quality control tests, 7 microsatellite markers can be used for the study

of the CPB populations, and 11 markers for the LFB wherein the 7 markers that were selected for CPB are also usable in the LFB. Most loci were affected by null alleles, while SAD and stuttering were not evidenced at any loci. Locus Cpb62 was found diagnostic between the two species and will be useful as additional tool in case of uncertain morphological determinations.

Effective population sizes varied from 50 to 4200 individuals for CPB from Indonesia, Malaysia and Philippines, and between 20 and 200 for LFB from Vietnam. Whether these differences are significant will require more subsamples of both species. Dispersal distances could be estimated from subdivision measures between sites occupied by the cocoa moth populations. Using this limited sampling effort, it seemed that this species can disperse over quite long distances, i.e. on average 17 km per generation, and with minimum and maximum distances of 9 and 31 km per generation, respectively. In view that a generation lasts only one month, dispersal over a twelve month period can be very large. Despite the potential large dispersal capabilities of CPB, no hybrids with LFB could be detected in areas where both crops are cultivated in close proximity. However, the large dispesal capacity observed in our analysis might also be due to human induced involvement of infested fruit with larvae between coutries/islands. This however will require more sampling of both species in closely located cacao and lychee plantations.

The results indicate the effectiveness of the selected loci to explore differences between CPB populations. A previous study analyzed the genetic structure of the CPB from Malaysia using mitochondrial DNA markers, and indicated that this species might have been exposed to a bottleneck event [15]. The availability of microsatellite loci is expected to provide more thorough and in-depth analysis that might explore more genetic differences in the different populations. Determining the level of gene flow between CPB populations in neighbouring countries might have economic consequences with respect to trading regulations and might help to develop sustainable AW-IPM strategies at the regional level.

## Supporting information

**S1 Table. Selected primers of the cocoa pod borer *Conopomorpha cramerella* microsatellites.** The primers were selected using msatcommander software with the partial genome available in the database, primers predicted to amplify a product length ranged between 180–300 nucleotides with minimum 11 repeat of each motif were selected.
(XLSX)

**S2 Table. List of primers successfully amplify PCR the expected PCR product with the DNA samples of the cocoa pod borer *Conopomorpha cramerella* and the litchi fruit borer *Conopomorpha sinensis*.** *: primers did not amplify PCR product in litchi fruit borer Conopomorpha sinensis samples collected from Vietnam.
(XLSX)

**S3 Table. List of genotypes (alleles) per locus for the cocoa pod borer *Conopomorpha cramerella* and the lychee fruit borer *Conopomorpha sinensis*.** Dye-labeled PCR were analysed by fragment analyzer and the resulted data were read with Genemapper.
(CSV)

**S4 Table. Comparison of Weir and Cockerham's unbiased estimator of Wright's $F_{IS}$ in males and females of using Wilcoxon signed rank test.**
(XLSX)

**S5 Table. Weir and Cockerham's unbiased estimator of Wright's $F_{ST}$ measured between females and males within each country and for each locus, and *G*-based randomization**

**test for significant of divergence.**
(XLSX)

**S6 Table. Null allele frequency estimates.**
(XLSX)

**S1 Fig.** (A) Wright's FIS with p-values for panmixia, null alleles, SAD, stuttering and null alleles and (B) FST with p-values for subdivision of the seven selected microsatellite loci of cocoa pod borer Conopomorpha cramerella. [•]: FIS or FST per locus, [x]: FIS per population [-]: 95% CI.
(TIF)

**S2 Fig. Correlation between the $F_{IS}$ and $F_{ST}$ values and the missing genotypes ($N_{Blanks}$) of cocoa pod borer *Conopomorpha cramerella*.** The correlation shown is after assuming that the missing data for Cpb122 and Cpb62 are not null homozygotes.
(TIF)

**S3 Fig. Correlation between the $F_{IS}$ values and the missing genotypes ($N_{Blanks}$) of lychee fruit borer *Conopomorpha sinensis*.**
(TIF)

**S1 File. R Markdowns.**
(DOCX)

**S2 File. Inclusivity in global research document.**
(PDF)

# Acknowledgments

This study was financed by the International Atomic Energy Agency (IAEA), Austria.

# Author Contributions

**Conceptualization:** Marc J. B. Vreysen, Adly M. M. Abd-Alla.

**Data curation:** Marynold Purificacion, Roslina Binti Mohd Shah, Saripah Binti Bakar, Anisah Bintil Savantil, Meriam Mohd Yusof, Divina Amalin, Hien Nguyen, Endang Sulistyowati, Aris Budiman, Arni Ekayanti, Jerome Niogret, Adly M. M. Abd-Alla.

**Formal analysis:** Marynold Purificacion, Thierry De Meeûs, Sophie Ravel, Adly M. M. Abd-Alla.

**Funding acquisition:** Marc J. B. Vreysen.

**Investigation:** Marynold Purificacion, Roslina Binti Mohd Shah.

**Methodology:** Marynold Purificacion, Roslina Binti Mohd Shah, Sophie Ravel, Adly M. M. Abd-Alla.

**Project administration:** Marc J. B. Vreysen.

**Resources:** Saripah Binti Bakar, Anisah Bintil Savantil, Meriam Mohd Yusof, Divina Amalin, Hien Nguyen, Endang Sulistyowati, Aris Budiman, Arni Ekayanti, Jerome Niogret, Adly M. M. Abd-Alla.

**Software:** Sophie Ravel, Adly M. M. Abd-Alla.

**Validation:** Thierry De Meeûs, Adly M. M. Abd-Alla.

**Visualization:** Marynold Purificacion, Thierry De Meeûs, Adly M. M. Abd-Alla.

**Writing – original draft:** Adly M. M. Abd-Alla.

**Writing – review & editing:** Roslina Binti Mohd Shah, Saripah Binti Bakar, Anisah Bintil Savantil, Meriam Mohd Yusof, Divina Amalin, Hien Nguyen, Endang Sulistyowati, Aris Budiman, Arni Ekayanti, Jerome Niogret, Sophie Ravel, Marc J. B. Vreysen.

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
