## [Decision Letter · Decision Letter 0]

1 Sep 2023

PONE-D-23-19428Development and characterization of microsatellite markers for population genetics of the cocoa pod borer Conopomorpha cramerella (Snellen) (Lepidoptera: Gracillaridae)PLOS ONE

Dear Dr. Abd-Alla,

Thank you for submitting your manuscript to PLOS ONE. After careful consideration, we feel that it has merit but does not fully meet PLOS ONE’s publication criteria as it currently stands. Therefore, we invite you to submit a revised version of the manuscript that addresses the points raised during the review process.

We look forward to receiving your revised manuscript.

Kind regards,

Wosene Gebreselassie Abtew, Ph.D.

Academic Editor

PLOS ONE

Journal Requirements:

4. We note that you have referenced (De Meeûs T, Noûs C. Effective population size of dioecious populations: some little improvements and detailed demonstrations. Revised version (V3) Submitted. 2022. doi:10.5281/zenodo.7143026) which has currently not yet been accepted for publication. Please remove this from your References and amend this to state in the body of your manuscript: (ie “Bewick et al. [Unpublished]”) as detailed online in our guide for authors

5. We note that Figure 1 in your submission contain map/satellite images which may be copyrighted. All PLOS content is published under the Creative Commons Attribution License (CC BY 4.0), which means that the manuscript, images, and Supporting Information files will be freely available online, and any third party is permitted to access, download, copy, distribute, and use these materials in any way, even commercially, with proper attribution. For these reasons, we cannot publish previously copyrighted maps or satellite images created using proprietary data, such as Google software (Google Maps, Street View, and Earth). For more information, see our copyright guidelines: http://journals.plos.org/plosone/s/licenses-and-copyright.

Additional Editor Comments:

Dear Dr. Abd-Alla, Thank you very much for considering Plos One to publish your research results. Your present manuscript has been reviewed by experts and recommended for minor revision. I have decided for major revision after looking at the comments and concerns of the reviewers. Please, take into account the enclosed reviewers’ comments and modify your manuscript according to their suggestions. Also provide justifications for the concerns of the first reviewer mentioned as major. It is important to use ink in different colors to highlight the changes performed in the revised version of your manuscript, so that it could be easy for the reviewers to find the amendments performed in the second review round. Along with your revised manuscript, you will need to supply a covering letter in which you list all the changes you have made to the manuscript, and in which you detail your responses to all the comments passed by the reviewers and the Editor. Should you disagree with any comment (s), please explain why.

Reviewers' comments:

Reviewer's Responses to Questions

**Comments to the Author**

1. Is the manuscript technically sound, and do the data support the conclusions?

Reviewer #1: Yes

Reviewer #2: Partly

2. Has the statistical analysis been performed appropriately and rigorously? 

Reviewer #1: Yes

Reviewer #2: I Don't Know

3. Have the authors made all data underlying the findings in their manuscript fully available?

Reviewer #1: Yes

Reviewer #2: Yes

4. Is the manuscript presented in an intelligible fashion and written in standard English?

Reviewer #1: Yes

Reviewer #2: No

5. Review Comments to the Author

Reviewer #1: The paper is about the cocoa pod borer (CPB), which is a major insect pest of cocoa production in Southeast Asia. CPB infests cocoa, rambutan, and langsat. The larvae of CPB enter the cocoa pods, which contain the beans used for chocolate production. On average, 40-60% of the cocoa beans produced in Southeast Asia are lost due to CPB, and up to 80% of the crop is lost in farms in the absence of CPB management. The paper aims to develop and characterize microsatellite markers of C. cramerella that can be used in further population genetics studies. The availability of these novel microsatellite loci will provide useful tools for the analysis of the population genetics and gene flow of these pests, to select suitable CPB strains to implement the Sterile Insect Technique (SIT).

The contributions of the manuscript towards the field

• The development and characterization of microsatellite markers for population genetics of the cocoa pod borer (CPB) Conopomorpha cramerella (Snellen) (Lepidoptera: Gracillaridae).

• The identification of suitable microsatellite loci in the genome of CPB that is partially sequenced.

• The analysis of moths collected from Indonesia, Malaysia, and the Philippines using the selected microsatellites, which helped in distinguishing differences between CPB populations from different locations.

• The testing of the selected microsatellites against a closely related species, i.e. the lychee fruit borer Conopomorpha sinensis (LFB) from Vietnam, which helped in identifying suitable loci for LFB.

• The availability of these novel microsatellite loci will provide useful tools for the analysis of the population genetics and gene flow of these pests, to select suitable CPB strains to implement the Sterile Insect Technique (SIT).

• The practical implications of this paper include the development of more effective and sustainable control strategies for CPB and LFB, which are serious pests of cocoa and lychee production, respectively.

Major comments:

• The sample size used in this study may not be representative of the entire population of CPB and LFB in the entire world. The explanation of why the three countries were chosen as sample sites were also not clear.

• The microsatellite markers developed in this study may not be applicable to other populations of CPB and LFB from different geographical locations. Are there enough data generated to counter this argument?

Minor comments:

• The phylogenetics analysis methods were not clearly explained. Which evolutionary model was used and why? What were the bootstrapping parameters used? Why NJ tree was chosen and not the maximum likelihood tree?

• Many of the acronyms did not have the full words? Eg CI, LD, SAD, aw-ipm etc. Yes, to a population geneticist they might be trivial but the author should make the manuscript easy to read by a wide audience.

• Some of the clickable hyperlinks are broken, eg Reference No. 3

• The figures do not have a high resolution. Please improve the resolution.

Reviewer #2: The development of microsatellite markers for Conopomorpha cramerella and Conopomorpha sinensis is a great advance in the management of these lepidopteran pests. As the authors rightly point out, this will deliver in-depth understanding of dispersal abilities and population structures and therefore better management strategies. I am an end user of microsatellites but have never developed any myself, so I am unsure about my suitability in assessing if these markers have been developed well and are fit for purpose. It seems that way given the array of statistical methods applied. That being said, I found the manuscript difficult to understand. Some sections are poorly written, particularly in the material and methods section.

As a summary I found the following points need to be addressed:

• It is unclear which markers are used for which Conopomorpha species. How many and which ones are shared?

• It is unclear if alleles are different between males and females for both species.

• It is unclear how were the selected loci being affected by null alleles (this is important to explain well as null alleles could invalidate the results).

• It is unclear which loci is to be used with which fluorescent dye. This is critical if I was to try these loci in my own experiments.

• It is unclear if allele sizes include or exclude the M13 adapter.

• How were both species identified?

Additional comments below:

Line 65 …..it infests…..

Line 66 no bracket after Bennet

Line 66 change ‘ovipositing’ for ‘oviposit’ and delete …’the eggs’ as oviposit means laying eggs.

Line 67 add after …’the pods…’ what happens when larvae enter the pods i.e., what kind of damage they do.

Line 70 delete the bracket before Bradley

Line 71 add ‘lepidopteran’ before ‘borer’

Line 75 change ‘i.e.’, for e.g., change ‘it’ for they, change ‘and’ for or (in labor-intensive and costly). Check grammar (used before the first of two (or occasionally more) given alternatives (the other being introduced by ‘or’, e.g., "either I accompany you to the classroom or I wait here")

Line 75 delete space after ‘bags’ and before the comma.

Line 76 I do not understand the meaning of ‘ineffective or viable’. Reword or delete.

Line 77 delete the ‘and’ and the bracket after ‘pheromones’.

Line 78 I do not understand the meaning of ‘or it has environmental concerns’. Reword.

Line 74-78 The sentence is too long. The authors should try to split this sentence into two.

Line 79 typo in autocidal

Line 89 change to ‘between individuals and within and between populations.’ One cannot assess the genetic variation within an individual!

Line 91-94 Add references and examples of invertebrate studies that used the molecular markers listed. What was achieved in those studies?

Line 92 a bracket is missing before COI.

Line 98-99 COI is not an ideal marker for population genetics even though it has been used, but this is mostly due to the lack of other better suited markers.

Line 96 ‘species specific’? not always. Some loci are shared between closely related species.

Line 98 change ‘the’ before ‘Conopomorpha for ‘any of the known…’

Line 107 delete ‘three Asian countries, including’

Line 109 I am not familiar with the ‘cylindrical plastic bag’ collecting method. Is it the same as net sweeping?

Line 110 …’infected cacao pods’. Does this mean that some individuals were larvae? add ‘and adults were trapped…’

Line 111 Are these insects adults or larvae?

Line 112 change ‘by’ for with

Line 114 change ‘by’ for with

Line 115 Was the whole specimen used for DNA extraction? How was this done i.e., using a mixer mill, a pestle?

Line 130 delete the bracket after ‘DNA’

Line 135-136 were the 12 primer pairs selected on the basis that they amplified consistently across all individuals? Or based on size? What was the final criteria used?

Line 396 which 11 loci out of 12 listed in Table 2 were preselected? Why then listing 12? Perhaps add a column in Table 2 explaining which loci are used for CPB and/or for LFB. This is not clear.

Line 396 …’seemed to work well…’ I am concern about this statement as I am not sure what it means. It either works or it doesn’t work!

Line 398 ‘most loci were affected by null alleles’…in that case

Line 400 why using microsatellites to differentiate between species when dissection of genitalia (in adults) and/or the COI marker (for all stages) can do this perfectly

Line 406 disperse

Line 407-408 Add the references that support such statements of distance dispersal.

Line 412 change …’the transfer (legal and illegal)…’ for …’human induced movement of…’

Table 1 How where the males and females identified? This was not explained in the Materials and Methods section.

6. PLOS authors have the option to publish the peer review history of their article (what does this mean?). If published, this will include your full peer review and any attached files.

Reviewer #1: No

Reviewer #2: No

---

## [Author Response · Author response to Decision Letter 0]

3 Oct 2023

Point-by-point response to Reviewers’ comments on the manuscript

PONE-D-23-19428

Development and characterization of microsatellite markers for population genetics of the cocoa pod borer Conopomorpha cramerella (Snellen) (Lepidoptera: Gracillaridae)

Marynold Purificacion, Roslina Binti Mohd Shah, Thierry De Meeûs, Saripah Binti Bakar4 Anisah Bintil Savantil, Meriam Mohd Yusof, Divina Amalin, Hien Nguyen, Endang Sulistyowati, Aris Budiman, Arni Ekayanti, Jerome Niogret, Sophie Ravel, Marc J.B. Vreysen, Adly M.M. Abd-Alla

Journal Requirements:

Author response: Manuscript has been revised based on PLOS ONE’s style requirements.

Please include a complete copy of PLOS’ questionnaire on inclusivity in global research in your revised manuscript.

Author response: The questionnaire on inclusivity in global research has been filled and attached to the manuscript as S2 file. In addition the following section was added to the materials and method section (lines 293-295)

“Inclusivity in global research:

Additional information regarding the ethical, cultural, and scientific considerations specific to inclusivity in global research is included in the Supporting Information (S2 file).

We note that you have stated that you will provide repository information for your data at acceptance. Should your manuscript be accepted for publication, we will hold it until you provide the relevant accession numbers or DOIs necessary to access your data. If you wish to make changes to your Data Availability statement, please describe these changes in your cover letter and we will update your Data Availability statement to reflect the information you provide.

Author response: The link of the data repository is tested and it is functional. No change in the data availability is needed

We note that you have referenced (De Meeûs T, Noûs C. Effective population size of dioecious populations: some little improvements and detailed demonstrations. Revised version (V3) Submitted. 2022. doi:10.5281/zenodo.7143026) which has currently not yet been accepted for publication. Please remove this from your References and amend this to state in the body of your manuscript: (ie “Bewick et al. [Unpublished]”) as detailed online in our guide for authors

Author response: The paper has now been published and the full reference is:

De Meeûs T, Noûs C (2023) A new and almost perfectly accurate approximation of the eigenvalue effective population size of a dioecious population: comparisons with other estimates and detailed proofs Peer Community Journal, 3, e51. https://doi.org/10.24072/pcjournal.280

We note that Figure 1 in your submission contain map/satellite images which may be copyrighted. All PLOS content is published under the Creative Commons Attribution License (CC BY 4.0), which means that the manuscript, images, and Supporting Information files will be freely available online, and any third party is permitted to access, download, copy, distribute, and use these materials in any way, even commercially, with proper attribution. 

We require you to either (1) present written permission from the copyright holder to publish these figures specifically under the CC BY 4.0 license, or (2) remove the figures from your submission or supply a replacement figure that complies with the CC BY 4.0 license.

Author response: Fig 1 was prepared using the batchgeo tool available online (https://batchgeo.com/) and the legend of Fig 1 was amended to indicate these details. As the figure is prepared by the authors, now permission is needed

PLOS authors have the option to publish the peer review history of their article (what does this mean?). If published, this will include your full peer review and any attached files.

Author response: We would like to publish the peer review history of the paper. 

Additional Editor Comments:

Dear Dr. Abd-Alla, Thank you very much for considering Plos One to publish your research results. Your present manuscript has been reviewed by experts and recommended for minor revision. I have decided for major revision after looking at the comments and concerns of the reviewers. Please, take into account the enclosed reviewers’ comments and modify your manuscript according to their suggestions. Also provide justifications for the concerns of the first reviewer mentioned as major. It is important to use ink in different colors to highlight the changes performed in the revised version of your manuscript, so that it could be easy for the reviewers to find the amendments performed in the second review round. Along with your revised manuscript, you will need to supply a covering letter in which you list all the changes you have made to the manuscript, and in which you detail your responses to all the comments passed by the reviewers and the Editor. Should you disagree with any comment (s), please explain why.

Author response: We thank the editor and the reviewers for their valuable comments which helped to improve the manuscript. We have revised the manuscript according to the editor and reviewers‘ suggestions. Our response to the reviewers’ comments is detailed below. The line number mentioned in our response refers to the line number of the revised manuscript with track changes.

Reviewers’ Comments:

Reviewer #1:

The paper is about the cocoa pod borer (CPB), which is a major insect pest of cocoa production in Southeast Asia. CPB infests cocoa, rambutan, and langsat. The larvae of CPB enter the cocoa pods, which contain the beans used for chocolate production. On average, 40-60% of the cocoa beans produced in Southeast Asia are lost due to CPB, and up to 80% of the crop is lost in farms in the absence of CPB management. The paper aims to develop and characterize microsatellite markers of C. cramerella that can be used in further population genetics studies. The availability of these novel microsatellite loci will provide useful tools for the analysis of the population genetics and gene flow of these pests, to select suitable CPB strains to implement the Sterile Insect Technique (SIT).

The contributions of the manuscript towards the field.

• The development and characterization of microsatellite markers for population genetics of the cocoa pod borer (CPB) Conopomorpha cramerella (Snellen) (Lepidoptera: Gracillaridae).

• The identification of suitable microsatellite loci in the genome of CPB that is partially sequenced.

• The analysis of moths collected from Indonesia, Malaysia, and the Philippines using the selected microsatellites, which helped in distinguishing differences between CPB populations from different locations.

• The testing of the selected microsatellites against a closely related species, i.e. the lychee fruit borer Conopomorpha sinensis (LFB) from Vietnam, which helped in identifying suitable loci for LFB.

• The availability of these novel microsatellite loci will provide useful tools for the analysis of the population genetics and gene flow of these pests, to select suitable CPB strains to implement the Sterile Insect Technique (SIT).

• The practical implications of this paper include the development of more effective and sustainable control strategies for CPB and LFB, which are serious pests of cocoa and lychee production, respectively.

Major comments:

The sample size used in this study may not be representative of the entire population of CPB and LFB in the entire world. The explanation of why the three countries were chosen as sample sites were also not clear.

Author response: While we agree with the reviewer that the sample size we used may not be representative of the entire population of CPB and LFB in the world, we believe that the result is still significant since this is the first set of microsatellites for the two species. In addition, the main objective of this paper is just to develop, characterize and validate the selected microsatellite markers that can be use in the future to analyze the population genetics of these insects. The deep analysis of the population genetics of these insects using the selected marker is currently on going in another study. 

The microsatellite markers developed in this study may not be applicable to other populations of CPB and LFB from different geographical locations. Are there enough data generated to counter this argument?

Author response: Although the number of tested population per country and the number of sample is rather low in this study, we believe that there is a high probability that the selected marker might work well with most of the populations of CPB and LFB from different geographical locations due to the following reasons: i) the geographical range of this preliminary population genetics study embraced almost all Southeast Asia; ii) The genetic differentiation within CPB was substantial enough with the seven loci kept after quality testing; iii) These seven loci also worked very well on the distant species LFB. Given these observations, there is no reason to doubt that other populations of the two species will be successfully studied, at least for the 7 loci in common between CFB and LFB. Moreover, The work is ongoing to select more markers to increase the resolution of the population genetics study and to cover all population of these two species

Minor comments:

The phylogenetics analysis methods were not clearly explained. Which evolutionary model was used and why? What were the bootstrapping parameters used? Why NJ tree was chosen and not the maximum likelihood tree?

Author response: We are not sure of what Referee 1 means by "evolutionary model" as we used the Neighbor Joining algorithm. This method was used following Saitou and Nei's and Takezaki and Nei's papers that concluded that Cavalli-Sforza and Edward's chord distance (D_CSE) based NJTree provided the most accurate trees. Moreover, using the INA algorithm of FreeNA (Chapuis and Estoup, 2007) allowed us to correct D_CSE for the presence of null alleles. Maximum-Likelihood, at least with Mega, requires sequence data.

We did not undertake any bootstrap procedure for this NJTree Construction. Nevertheless, given the level of significance found between the different subsamples of the CPB, and the unambiguous hiatus between CPB and LFB, we did not see the relevance of such an approach. As explained in the text, the purpose of this section was to obtain a visual and easy to interpret image of the genetic distances between individuals.

We have changed the sentence of line 223-229 into:

Lastly, to visualize the genetic relationships between each individuals, a genetic distance matrix was computed using Cavalli-Sforza and Edwards' chord distance (DCSE) (Cavalli-Sforza and Edwards, 1967), corrected for null alleles with the INA procedure with FreeNA (Chapuis and Estoup, 2007). This matrix was used to build a neighbor-joining tree (NJTree) (Saitou and Nei) with Mega 7 software (Kumar et al, 2016) as recommended by Takezaki and Nei. This version of MEGA indeed allow to import the resulting tree into an object that is editable with a drawing/presentation program (Impress, PowerPoint, etc.), while more recent versions only create an image that cannot be modified.

Please, note that Figure 3 was amended in agreement with the message sent by T. de Meeûs to the editor of PLoS 1 the 22/06/2023.

Many of the acronyms did not have the full words? Eg CI, LD, SAD, aw-ipm etc. Yes, to a population geneticist they might be trivial but the author should make the manuscript easy to read by a wide audience.

Author response: AW-IPM was defined to area-wide integrated pest management in Lines 112-113; LD to linkage disequilibrium in line 232; SAD was defined to short alleles dominance in line 250. we added the sentence: "We computed 95% confidence intervals (CI) with 5000 bootstraps over loci (lines 276-277).

Some of the clickable hyperlinks are broken, eg Reference No. 3

Author response: Hyperlinks are now working.

The figures do not have a high resolution. Please improve the resolution.

Author response: We uploaded the figures with a higher resolution.

Reviewer #2:

The development of microsatellite markers for Conopomorpha cramerella and Conopomorpha sinensis is a great advance in the management of these lepidopteran pests. As the authors rightly point out, this will deliver in-depth understanding of dispersal abilities and population structures and therefore better management strategies. I am an end user of microsatellites but have never developed any myself, so I am unsure about my suitability in assessing if these markers have been developed well and are fit for purpose. It seems that way given the array of statistical methods applied. That being said, I found the manuscript difficult to understand. Some sections are poorly written, particularly in the material and methods section.

As a summary I found the following points need to be addressed:

• It is unclear which markers are used for which Conopomorpha species. How many and which ones are shared?

Author response: . All markers were tested on the two species (Lines 133-135). These loci are also clearly nominated in Tables and Figures. Nevertheless, we added in the discussion that all seven loci that were selected after quality testing in CPB are also usable in the LFB (Line 461-462).

• It is unclear if alleles are different between males and females for both species.

Author response: Al loci appeared to be autosomal. The comparison between males and females for both species are in Lines 335-339. Regarding alleles, we hardly see how alleles may differ between genders since at each generation, all zygotes are produced by half from female and half from a male genome.

• It is unclear how were the selected loci being affected by null alleles (this is important to explain well as null alleles could invalidate the results).

Author response: How null alleles affected the FIS is explained in the text and figures. We also have added the S6 Table providing null allele frequency estimates. We can notice that the averages are far from the 0.2 threshold described in Séré et al (2017), and that null alleles explained 80% and 75% of FIS variation, with an intercept (FIS with no null alleles) FIS_0=-0.03 in CPB and LFB, respectively. Given the number of subsamples and their sizes, this represents a good enough result. For genetic differentiation, we used FreeNA which is known to accurately correct for the effects of null alleles. The manuscript was amended to reflect these details in lines 394-400. 

• It is unclear which loci is to be used with which fluorescent dye. This is critical if I was to try these loci in my own experiments.

Author response: The manuscript was amended to add the suggested details (which dye was used for each microsatellite) in lines 185-186. 

• It is unclear if allele sizes include or exclude the M13 adapter.

Author response: Allele sizes include the M13 adapter. A footnote was added to Table 2 to indicate this. 

• How were both species identified?

Author response: CPB and LFB were identified by the adults morphology and the historical information about the insect distribution.

• Table 1 How where the males and females identified? This was not explained in the Materials and Methods section. 

Author response: Males and females were distinguished by examining the ventrocaudal segments of the abdomen, The manuscript was amended to indicate these details in line 152-153.

Additional comments:

Line 65 …..it infests….. 

Author response: Changed in the manuscript as suggested (Line 85)

Line 66 no bracket after Bennet

Author response: We did not remove the bracket as we are indicating the full scientific name of langsat (Lansium parasiticum (Osbeck) Sahni & Bennet) (Lines 86-87)

Line 66 change ‘ovipositing’ for ‘oviposit’ and delete …’the eggs’ as oviposit means laying eggs.

Line 67 add after …’the pods…’ what happens when larvae enter the pods i.e., what kind of damage they do. 

Author response: Sentence was changed to “CPB adults oviposit on the cocoa pods and the larvae enter the pods to feed on the pulp and placenta surrounding the beans that are used for chocolate production.” As suggested (Lines 87-89)

Line 70 delete the bracket before Bradley

Line 71 add ‘lepidopteran’ before ‘borer’

Author response: Changed as suggested (lines 91-92)

Line 75 change ‘i.e.’, for e.g., change ‘it’ for they, change ‘and’ for or (in labor-intensive and costly). Check grammar (used before the first of two (or occasionally more) given alternatives (the other being introduced by ‘or’, e.g., "either I accompany you to the classroom or I wait here")

Line 75 delete space after ‘bags’ and before the comma.

Line 76 I do not understand the meaning of ‘ineffective or viable’. Reword or delete.

Line 77 delete the ‘and’ and the bracket after ‘pheromones’.

Line 78 I do not understand the meaning of ‘or it has environmental concerns’. Reword.

Line 74-78 The sentence is too long. The authors should try to split this sentence into two.

Author response: Changed as suggested and the manuscripr was changes to have the following sentence “Various control measures have been implemented to manage populations of these pests however they are either labor-intensive and costly (e.g., sleeving the cocoa pods with plastic bags), ineffective (e.g., biological control methods such as the use of natural enemies and sex pheromones) or has a negative impact on the environment (e.g., use of chemical insecticide). (lines 95-98).

Line 79 typo in autocidal

Author response: Corrected in the manuscript as suggested (Line 99).

Line 89 change to ‘between individuals and within and between populations.’ One cannot assess the genetic variation within an individual!

Author response: The manuscript was Changed to “Such knowledge can be relatively easily assessed by the study of genetic variation of molecular markers between individuals, and within and between subsamples in space and time.” (122-124)

Line 91-94 Add references and examples of invertebrate studies that used the molecular markers listed. What was achieved in those studies?

Author response: References add as suggested (Examples are in Ref 12-13). We also added Ref 16 on using mitochondrial markers for population genetics of LFB. 

Line 92 a bracket is missing before COI.

Author response: Bracket added as suggested in Line 126, also mtDNA was added as abbreviation for mitochondrial DNA. 

Line 98-99 COI is not an ideal marker for population genetics even though it has been used, but this is mostly due to the lack of other better suited markers.

Author response: We agree with this comment. It emphasizes the need to have a more suitable marker like microsatellites for population genetics of CPB. 

Line 96 ‘species specific’? not always. Some loci are shared between closely related species.

Author response: ‘species-specific’ was removed. 

Line 98 change ‘the’ before ‘Conopomorpha for ‘any of the known…’

Author response: Changed to ‘any of the known Conopomorpha spp.’ As suggested (Line 132) and the sentence was changed to “However, no microsatellites had been developed for any of the known Conopomorpha spp., and earlier attempts to analyze their population structure using COI and EF-1� remained inconclusive” lines 131-133.

Line 107 delete ‘three Asian countries, including’

Author response: The sentence was changed to “Adult CPB moths were collected from Malaysia (Mal), Indonesia (Indo), and the Philippines (Phil)” lines 145-146

Line 109 I am not familiar with the ‘cylindrical plastic bag’ collecting method. Is it the same as net sweeping?

Author response: No, it is not the same.

Line 110 …’infected cacao pods’. Does this mean that some individuals were larvae? add ‘and adults were trapped…’

Author response: Yes, some larvae were also present in the infected cacao pods, but we only used the adults collected. 

Line 111 Are these insects adults or larvae?

Author response: All insects collected and used in this paper were adults. 

Line 112 change ‘by’ for with

Author response: Changed in the manuscript (Line 150) 

Line 114 change ‘by’ for with

Author response: Changed in the manuscript (Line 152) 

Line 115 Was the whole specimen used for DNA extraction? How was this done i.e., using a mixer mill, a pestle?

Author response: Whole specimen was used for the DNA extraction. Samples were grind in liquid nitrogen and the DNA was extracted as per manufacture’s instruction of the DNeasy® Blood & Tissue Kit as described in line 155-157 .

Line 130 delete the bracket after ‘DNA’

Author response: Bracket removed (Line 173) 

Line 135-136 were the 12 primer pairs selected on the basis that they amplified consistently across all individuals? Or based on size? What was the final criteria used?

Author response: The 12 primer pairs were selected based on consistent amplification across all individuals and expected fragment size, The manuscript was changed to indicate these details (Line 179-180)

Line 396 which 11 loci out of 12 listed in Table 2 were preselected? Why then listing 12? Perhaps add a column in Table 2 explaining which loci are used for CPB and/or for LFB. This is not clear.

Author response: Cpb136 was excluded due to amplification failures (see Line 337-338). All 12 microsatellites were tested on CPB and LFB. The footnote in table 2 was amended to clarify this point

Line 396 …’seemed to work well…’ I am concern about this statement as I am not sure what it means. It either works or it doesn’t work!

Author response: Statements with ‘seemed to work well’ changed to ‘worked well’ (Line 464)

Line 398 ‘most loci were affected by null alleles’…in that case

Author response: yes

Line 400 why using microsatellites to differentiate between species when dissection of genitalia (in adults) and/or the COI marker (for all stages) can do this perfectly

Author response: The manuscript was changed to indicate the microsatellites Cpb62 can be used as additional tools for species identification line 469.

Line 406 disperse

Author response: Corrected in the manuscript (Line 475)

Line 407-408 Add the references that support such statements of distance dispersal.

Author response: These statements are based on the data generated. The methodology for this is explained in Line 312-314. 

Line 412 change …’the transfer (legal and illegal)…’ for …’human induced movement of…’

Author response: Changed in the manuscript (Line 483)

---

## [Decision Letter · Decision Letter 1]

10 Jan 2024

Development and characterization of microsatellite markers for population genetics of the cocoa pod borer Conopomorpha cramerella (Snellen) (Lepidoptera: Gracillaridae)

PONE-D-23-19428R1

Dear Dr. Abd-Alla,

We’re pleased to inform you that your manuscript has been judged scientifically suitable for publication and will be formally accepted for publication once it meets all outstanding technical requirements.

Kind regards,

Jianhong Zhou

Staff Editor

PLOS ONE

Additional Editor Comments (optional):

Reviewers' comments:

Reviewer's Responses to Questions

**Comments to the Author**

1. If the authors have adequately addressed your comments raised in a previous round of review and you feel that this manuscript is now acceptable for publication, you may indicate that here to bypass the “Comments to the Author” section, enter your conflict of interest statement in the “Confidential to Editor” section, and submit your "Accept" recommendation.

Reviewer #1: All comments have been addressed

Reviewer #2: All comments have been addressed

2. Is the manuscript technically sound, and do the data support the conclusions?

Reviewer #1: Yes

Reviewer #2: Yes

3. Has the statistical analysis been performed appropriately and rigorously? 

Reviewer #1: Yes

Reviewer #2: Yes

4. Have the authors made all data underlying the findings in their manuscript fully available?

Reviewer #1: Yes

Reviewer #2: Yes

5. Is the manuscript presented in an intelligible fashion and written in standard English?

Reviewer #1: Yes

Reviewer #2: Yes

6. Review Comments to the Author

Reviewer #1: (No Response)

Reviewer #2: All my comments have been addressed appropriately. I recommend this manuscript to be accepted for publication.

7. PLOS authors have the option to publish the peer review history of their article (what does this mean?). If published, this will include your full peer review and any attached files.

Reviewer #1: No

Reviewer #2: No

---

## [Editor Report · Acceptance letter]

31 Jan 2024

PONE-D-23-19428R1 

PLOS ONE

Dear Dr. Abd-Alla, 

I'm pleased to inform you that your manuscript has been deemed suitable for publication in PLOS ONE. Congratulations! Your manuscript is now being handed over to our production team.

Kind regards, 

on behalf of

Dr. Jianhong Zhou 

Staff Editor

PLOS ONE